# Circadian Disruption Primes Myofibroblasts for Accelerated Activation as a Mechanism Underpinning Fibrotic Progression in Non-Alcoholic Fatty Liver Disease

**DOI:** 10.3390/cells12121582

**Published:** 2023-06-08

**Authors:** Elliot Jokl, Jessica Llewellyn, Kara Simpson, Oluwatobi Adegboye, James Pritchett, Leo Zeef, Ian Donaldson, Varinder S. Athwal, Huw Purssell, Oliver Street, Lucy Bennett, Indra Neil Guha, Neil A. Hanley, Qing-Jun Meng, Karen Piper Hanley

**Affiliations:** 1Wellcome Centre for Cell-Matrix Research, Faculty of Biology, Medicine & Health, Manchester Academic Health Science Centre, University of Manchester, Oxford Road, Manchester M13 9PL, UK; 2Division of Diabetes, Endocrinology and Gastroenterology, Faculty of Biology, Medicine & Health, Manchester Academic Health Science Centre, University of Manchester, Oxford Road, Manchester M13 9PL, UK; 3Department of Life Sciences, Manchester Metropolitan University, Manchester M15 6BH, UK; 4Bioinformatics Core Facility, Faculty of Life Sciences, University of Manchester, Manchester M13 9PL, UK; 5Manchester University NHS Foundation Trust, Oxford Road, Manchester M13 9WL, UK; 6National Institute for Health Research, Nottingham Biomedical Research Centre, Nottingham University Hospitals National Health Service Trust, University of Nottingham, Nottingham NG7 2RD, UK

**Keywords:** circadian rhythm, CLOCK, fibrosis, NAFLD, liver disease

## Abstract

Circadian rhythm governs many aspects of liver physiology and its disruption exacerbates chronic disease. CLOCKΔ19 mice disrupted circadian rhythm and spontaneously developed obesity and metabolic syndrome, a phenotype that parallels the progression of non-alcoholic fatty liver disease (NAFLD). NAFLD represents an increasing health burden with an estimated incidence of around 25% and is associated with an increased risk of progression towards inflammation, fibrosis and carcinomas. Excessive extracellular matrix deposition (fibrosis) is the key driver of chronic disease progression. However, little attention was paid to the impact of disrupted circadian rhythm in hepatic stellate cells (HSCs) which are the primary mediator of fibrotic ECM deposition. Here, we showed in vitro and in vivo that liver fibrosis is significantly increased when circadian rhythm is disrupted by CLOCK mutation. Quiescent HSCs from CLOCKΔ19 mice showed higher expression of RhoGDI pathway components and accelerated activation. Genes altered in this primed CLOCKΔ19 qHSC state may provide biomarkers for early liver disease detection, and include AOC3, which correlated with disease severity in patient serum samples. Integration of CLOCKΔ19 microarray data with ATAC-seq data from WT qHSCs suggested a potential CLOCK regulome promoting a quiescent state and downregulating genes involved in cell projection assembly. CLOCKΔ19 mice showed higher baseline COL1 deposition and significantly worse fibrotic injury after CCl_4_ treatment. Our data demonstrate that disruption to circadian rhythm primes HSCs towards an accelerated fibrotic response which worsens liver disease.

## 1. Introduction

Non-alcoholic fatty liver disease (NAFLD) is a major, increasing health care burden with an estimated incidence of 25% worldwide, increasing to 55% in those with type 2 diabetes [1]. NAFLD results from excessive accumulation of fat (or steatosis) in liver tissue, in the absence of significant alcohol intake, with primary risk factors including metabolic syndrome and obesity [2]. NAFLD progresses to non-alcoholic steatohepatitis (NASH) in 20–30% of cases, resulting in liver fibrosis (or scarring), cirrhosis and hepatocellular carcinoma (HCC). Despite increasing insight, the mechanisms promoting development of NASH remain incomplete.

Multiple factors lead to abnormal fat accumulation in hepatocytes and increasing evidence suggests that disruption of circadian rhythms, through Western lifestyle, are directly linked to NAFLD [3,4]. In humans, CLOCK (circadian locomotor output cycles kaput) gene variants and related haplotypes were associated with increased susceptibility to NAFLD and severity in disease progression [5] and behavioural disruption to circadian rhythm, for example shift work, also contribute to liver disease risk [6]. Similarly, in mouse models, CLOCK is closely associated with NAFLD. Loss of CLOCK results in reduced hepatic triglyceride accumulation following a high fat diet (HFD); whereas mice harbouring a CLOCK mutation (CLOCKΔ19) have dysregulated circadian rhythm, hepatic steatosis, obesity and metabolic syndrome [7,8,9]. Although these studies provide critical insight into dysregulated mechanisms associated with lipid metabolism in NAFLD, fibrosis is the main determinant of mortality in NASH.

Fibrosis, characterised by deposition of excessive extracellular matrix (ECM), is mediated by hepatic stellate cells (HSCs) which transition from a quiescent (qHSCs) to an activated (aHSCs; liver myofibroblasts) state in response to liver injury including NASH [10]. In the liver, approximately 10% of the transcriptome is thought to be rhythmically expressed by a 24 h intrinsic biological clock via interconnected transcriptional and translational feedback loops [11,12,13]. CLOCK is a core component of this transcriptional feedback loop and dimerises with BMAL1 (Brain and muscle-ARNT-like 1, also known as ARNTL1) to regulate downstream target genes. The CLOCKΔ19 mutant protein dimerises with BMAL1 but fails to achieve transcriptional regulation, thus acting in a dominant negative manner [14,15]. Oscillation in transcription is achieved by a negative feedback loop of downstream factors including Period (PER)/Cryptochrome (CRY), which repress the transcriptional activity of the CLOCK/BMAL complex [16]. Studies in HSCs indicated BMAL1 overexpression inhibits HSC activation and improves myofibroblast accumulation and fibrosis in vivo [17]. Moreover, studies in other organs suggest disruption of the circadian clock leads to progressive disease and, in some cases, spontaneous fibrosis [18]. These observations are suggestive of circadian control of HSC activation and ECM deposition.

In this study, we utilised CLOCKΔ19 mice as a model of NAFLD to examine the in vitro and in vivo impacts of disrupted circadian rhythm driving the fibrotic response toward NASH. In contrast to studies in lung [18], the liver of CLOCKΔ19 mutant animals did not show overt fibrosis. However, on closer inspection, we observed micro-fibrosis associated with increased collagen fibrils and ‘loosening’ of the sinusoidal space where qHSCs reside. In response to a fibrotic insult induced by 6 weeks-carbon tetrachloride (CCl_4_), CLOCKΔ19 mice displayed significant liver damage, steatosis and fibrosis reminiscent of NASH. Critically, we demonstrate that quiescent HSCs in CLOCKΔ19 appear to reside in a primed pro-fibrotic state that is likely responsible for the enhanced fibrotic phenotype observed following an insult. From genomic analysis, we identified several markers perturbed in CLOCKΔ19 qHSCs as potential markers of disease and show Amine Oxidase Copper containing 3 (AOC3) progressively tracks liver disease severity in patient serum samples. To provide mechanistic insight, we constructed a CLOCK regulome based on ATAC-seq data from qHSCs and aHSCs, cross-referencing with genes known to be perturbed in CLOCKΔ19 HSCs. Our data suggest that CLOCK acts to maintain a quiescent state, consistent with HSC activation correlating with circadian rhythm disruption. Taken together, these data provide insight into the importance of circadian clock in driving fibrosis from activated HSCs to explain aspects of NAFLD progression to NASH.

## 2. Materials and Methods

### 2.1. Mice and In Vivo Fibrosis

CLOCKΔ19 mice were gifted by Professor J. Takahashi at the University of Texas Southwestern Medical Centre [9]. Animals were kept under normal 12 h light/12 h dark schedule with *ad libitum* access to food and water. CLOCKΔ19 mice used in this study were homozygous mutants.

Liver fibrosis was induced in 8-week-old male mice by IP injection of CCl_4_ at a ratio of 1:3 *v*/*v* in olive oil (2 μL/g) twice a week for 6 weeks or olive oil only (controls). At harvesting, samples for histology were fixed in 4% paraformaldehyde (PFA) overnight followed by storage in 70% ethanol. Samples for RNA/protein analyses were snap frozen and kept at −80 degrees until further processing. Blood was also collected for liver function tests.

### 2.2. Hepatic Stellate Cell Isolation and In Vitro Culture

HSCs were isolated from whole livers of mice as previously described [19,20]. Briefly, under terminal anaesthesia the portal vein was cannulated to perfuse the liver with an enzyme mix of pronase and collagenase in HBSS. Livers were excised and mechanically disrupted with a razor blade before filtration through a 100 μm mesh to produce a cell suspension. After washing steps, HSCs were isolated using an Optiprep density gradient, washed and resuspended in stellate growth media (16% FBS, 1% pen/strep in high glucose DMEM enriched with L-glutamine).

To analyse circadian rhythm in gene expression, cells were synchronised using a serum shock. When cells reached approximately 70% confluency after in vitro activation, media were changed to DMEM supplemented with 50% FBS for 2 h, then replaced by normal 16% FBS growth media. Circadian time of 0 (T0) was taken 24 h after the serum shock. Cells were harvested using RIPA (protein) or RLT (RNA) over 48 h every 4 h.

To assess rhythmicity of cells, real time bioluminescence was recorded using a LumiCycle apparatus (ActiMetrics, Wilmette, IL, USA) on primary HSCs isolated from PER2::LUC mice [21]. Cells were grown to 70% confluency before being synchronised using dexamethasone (10 nM, Sigma, Gillingham, UK). Media on cells were changed to sterile filtered recording media (20 mM glucose, 10 mM HEPES, 4 mM NaHCO_3_, 5% FBS and 1 mM luciferin), and dishes sealed with petroleum jelly and a glass coverslip. Baseline subtraction was carried out using a 24 h moving average.

### 2.3. Histology and Western Blot

Protein from cells and tissues was extracted in RIPA buffer and protein concentration determined by BCA assay to allow equivalent loading between samples. Expression levels were determined by a standard Western blotting protocol using anti-SOX9 (Millipore, Watford, UK, AB5535; 1:5000), aSMA (DAKO, Cheshire, UK, Z0097; 1:100), CLOCK (Abcam, Cambridge, UK, ab3517; 1:1000) and COL1A1 (Southern Biotech, Birmingham, AL, USA, 1310-01; 1:100) primary antibodies and corresponding species-specific HRP conjugated secondary antibodies. β-actin HRP (Sigma, Gillingham, UK, A2228; 1:50,000) was used as a loading control for normalisation.

Fixed tissue was embedded in paraffin wax and sectioned at 5 μm thickness. Picrosirius Red staining, immunohistochemistry and quantification was performed as previously described using anti- aSMA (DAKO, Z0097; 1:100), F4/80 (Abcam, ab6640, 1:500), COL1A1 (Southern Biotech, 1310-01; 1:100) primary antibodies, species-specific biotinylated secondary antibodies (Vector, Newark, CA, USA) and streptavidin-HRP (Vector, SA-5004; 1:200).

### 2.4. Microarray Analysis

A microarray was performed on RNA from HSCs isolated from CLOCKΔ19 mice and wild type (WT) controls. RNA was extracted using a RNeasy purification kit (Qiagen, Manchester, UK). Three samples per mouse were taken: quiescent HSCs, activated HSCs at T0 and activated HSCs at T12. An Affymetrix GeneChip Mouse Genome 430 2.0 Array was used allowing for the analysis of 39,000 transcripts. Principle component analysis was used to compare variability between sample batches (*n* = 3). Ratio, Fold-change, log2ratio, *p*-values and false discovery rates (fdr) were also calculated for comparison of samples. Results were further analysed using Ingenuity pathway analysis (Qiagen) to identify biological pathways associated with observed changes between samples.

### 2.5. ATAC-Seq Integration

ATAC-seq data set was obtained as per [22]. Peaks containing a CLOCK-binding motif were identified using simple enrichment analysis. Peaks with a >2-fold concentration enrichment in qHSCs/aHSCs (or vice versa) were selected and the nearest gene identified using RnaChipIntegrator v2.0.0 (https://github.com/fls-bioinformatics-core/RnaChipIntegrator). To validate a regulatory relationship, this list of genes was filtered to select only genes that were >2-fold up- or down-regulated in CLOCKΔ19 in the respective quiescent or activated state. EnrichR (https://maayanlab.cloud/Enrichr) [23] was used to identify significantly enriched GO Biological Process terms.

### 2.6. Human Serum Analysis

Patients were recruited to the ID-LIVER project prospectively through Liver Assessment Clinics at Manchester University NHS Foundation Trust hospitals since October 2020. Ethical approval was granted by REC North of Scotland (IRAS ID: 273633; REC reference: 20/NS/0055). Patients were either referred by primary care providers to secondary care specialists for fibrosis assessment following abnormal liver biochemistry, abnormal liver imaging, or identified in the community as having risk factors for liver disease. After informed consent, all patients underwent routine non-invasive clinical investigations to determine fibrosis severity including liver stiffness measurements (LSM) using vibration controlled transient elastography (FibroScan, Echosens, Paris, France) and blood-based liver fibrosis scoring systems (FIB-4 score [24] and NAFLD fibrosis score [25]). AOC3 was quantified using a DuoSet VAP1/AOC3 Elisa kit (DY3957, Bio-Techne, Abingdon, UK) and ancillary reagents following manufacturer’s instructions with serum diluted at 1:1000.

The cohort tested comprised of 102 male patients and 94 female patients. The mean age was 56.1 years. 134 (68.4%) patients had non-alcoholic fatty liver disease (NAFLD). 24 (12.2%) patients had alcohol-related liver disease and 20 (10.2%) patients had a combination of metabolic and alcoholic-related liver disease. A total of 12 (6.1%) patients did not have any evidence of liver disease following investigations, 2 (1.0%) patients had autoimmune disease, 2 (1.0%) patients were found to have gallstones causing abnormal liver transaminases and 1 (0.5%) patient had a drug induced liver injury. The cause of disease was unclear in 1 (0.5%) patient. The mean BMI of the cohort was 32.5 kg/m^2^. An amount of 41 (20.9%) patients had type 2 diabetes. The median alcohol consumption across the whole cohort was 1 unit per week. A total of 50 (25.5%) patients had a LSM > 8.0 kPa, indicating significant fibrosis and 31 (15.8%) patients had a LSM > 15.0 kPa indicating liver cirrhosis. AOC3 data were separately compared against LSM with patients stratified into three groups (<8 kPa; 8–15 kPa; >15 kPa), and against FIB-4 score with patients were stratified into three groups based on British Society of Gastroenterology guidelines for the management of abnormal LFTs (<1.3; 1.3–3.2; >3.2) [26].

### 2.7. Statistical Analyses

Statistical analyses were performed in Graphpad Prism 9 (Graphpad Software, Boston, MA, USA) using the tests indicated in the figure legends. Data were assessed for normality/lognormality using the Shapiro–Wilk test to determine the use of parametric or nonparametric statistical tests.

## 3. Results

### 3.1. CLOCKΔ19 Mice Have an Enhanced Fibrotic Response to CCl_4_ Treatment

Previous studies reported spontaneous lung fibrosis in CLOCKΔ19 mice. We, therefore, looked for a parallel phenotype of spontaneous liver fibrosis. Histological comparisons between the livers of naïve WT and CLOCKΔ19 mice showed an increase in parenchymal COL1 staining in the mutant, with elevated COL1 confirmed by Western blot (Figure 1A,B). However, no increase was observed in other pro-fibrotic markers such as α-SMA or SOX9 (Figure 1B,C). This suggests naive CLOCKΔ19 mice may have a higher baseline of interstitial collagen deposition, but fall short of full, spontaneous fibrosis in the liver.

To investigate the effect of a fibrotic insult on these mice, fibrosis was induced by 6 weeks of treatment with CCl_4_. After CCl_4_, a significant increase in scarring was observed in the CLOCKΔ19 tissue as well as double the number of portal-to-portal tract bridges formed by the scar relative to WT (Figure 2A–C). Increased a-SMA staining also indicates more active myofibroblasts in the CLOCKΔ19 livers (Figure 2A,D). No difference in relative liver weights were observed between WT and CLOCKΔ19 mice. Similarly, although elevated ALT was detected in the serum of CCl_4_-treated animals relative to controls, no significant differences were observed between WT and CLOCKΔ19 mice (Appendix A).

These data generated the hypothesis that HSCs in the CLOCKΔ19 mice may exist in a primed state which accelerates fibrosis upon injury.

### 3.2. Disruption to Circadian Rhythm Accelerates HSC Activation

Firstly, we sought to establish whether HSCs show circadian rhythmicity. PER2::LUC bioluminescence was measured in synchronised cells over 7 days. A rhythm of approximately 23.5 h was observed and maintained for 4 days in culture (Figure 3A). This circadian expression of an endogenous clock gene was confirmed through qPCR of CRY1, a core clock gene, over a 48 h time course of synchronised mHSCs (Figure 3B). To explore the effects of impaired circadian rhythm on HSCs, we carried out a time course of activation in culture. Significantly, CLOCKΔ19 HSCs displayed earlier activation and express profibrotic markers at day 5, whereas WT cells only express these from day 7 (Figure 3C), whilst proliferation and cell viability remained unchanged (Appendix A).

To understand the mechanisms underlying the response of HSCs following impaired CLOCK signalling, microarray analysis was performed on quiescent and 10-day-activated HSCs, synchronised and collected either 24 or 36 h later. Analysis included clustering of differentially expressed genes into nine categories (Figure 4A). In cluster 4, 79 genes were up regulated in CLOCKΔ19 qHSCs compared to WT qHSCs. Moreover, these genes were also up regulated in both WT and CLOCKΔ19 aHSCs independent of time collected; indicative of their profibrotic nature. In cluster 6, 275 genes were down regulated in CLOCKΔ19 qHSCs when compared to WT qHSCs and also down regulated in both WT and CLOCKΔ19 aHSCs regardless of time collected. Gene lists from these two clusters were analysed using ingenuity pathway analysis and the top 10 canonical pathways were identified. The top pathway that was increased in cluster 4 was hepatic fibrosis/hepatic Stellate cell activation (Figure 4B). Additionally, of interest is RhoGDI signalling due to its role in cell proliferation, apoptosis, differentiation, migration, cytoskeletal reorganisation, and membrane trafficking. In cluster 6, we noticed pathways classically associated with a quiescent HSC phenotype were downregulated in activated cells. Moreover, these pathways were similarly reduced in CLOCKΔ19 quiescent HSCs compared to WT (4C). The top pathway was farnesoid X receptor (FXR)/retinoid X receptor (RXR) activation, a pathway classically linked to the control of metabolic pathways including bile, lipid and glucose metabolism. A second nuclear receptor pathway which was downregulated in these cells when compared to WT quiescent HSCs was pregnane X receptor (PXR)/RXR activation, a pathway involved in drug metabolism. Central to both these pathways is PPARα, also significantly reduced in quiescent CLOCKΔ19 HSCs. Taken together, these data suggest that circadian disruption causes HSCs to have an altered metabolic state that potentially primes cells for profibrotic activation. We also examined the expression of key circadian rhythm genes between experimental groups in our microarray (Appendix A) but found few significant changes beyond reduction in *Per1* and *Rorγ* in aHSCs compared to qHSCs regardless of mutation status. The absence of more changes may reflect the loss of robust circadian rhythm due to desynchronization in culture which might not be fully restored by serum shock.

### 3.3. Factors Priming CLOCKΔ19 HSCs Include Liver Disease Biomarkers

To capture additional factors underlying the primed phenotype of CLOCKΔ19 qHSCs, including those that might not directly correlate to activation in our clustering analyses, we selected all genes that were over 1.5 fold up- or downregulated in CLOCKΔ19 vs. WT qHSCs (*p* < 0.1). Mirroring the previous clustering analyses, genes upregulated in CLOCKΔ19 qHSCs were enriched for extracellular matrix organisation, and downregulated genes included many metabolic pathways associated with quiescence including fatty acid oxidation and retinoid metabolism (Figure 5A).

To determine potential biomarkers for early liver disease, we turned our attention to genes upregulated in CLOCKΔ19 qHSCs which were annotated to the extracellular space GO component, reasoning that changes in these factors are more likely to be measurable in patient serum (Appendix A). The interrogation of the literature highlighted a subset of these genes as known correlates to liver disease severity (Figure 5B).

AOC3 is one such gene which is upregulated in CLOCKΔ19 qHSCs and was previously shown to correlate to human liver disease [27,28]. We sought to explore whether higher serum AOC3 might be an “early warning sign” of human liver fibrosis, given it’s upregulation in primed HSCs. Based on transient elastography, our analysis showed a progressive upward trend in serum AOC3 between non-fibrotic, fibrotic and cirrhotic patients of mixed aetiology, though this was only able to significantly differentiate between cirrhotic patients and the other two groups (Figure 5(Ci)). Contrasting AOC3 to FIB-4 score showed significant discrimination between low, intermediate and high FIB-4 groups (Figure 5(Cii)). This suggests that elevated AOC3 may act as an early marker of liver disease.

### 3.4. In Silico Analysis Predicts Direct Regulation of Quiescence Associated Genes by CLOCK

To explore which genes associated with primed HSCs might be directly regulated by CLOCK itself, we utilised ATAC-seq data from an activation time-course of WT HSCs. Quiescence-associated peaks were identified by twofold upregulation of peak volume in qHSCs compared to aHSCs, and activation-associated peaks were identified inversely. Simple enrichment analysis identified that the CLOCK motif was significantly enriched in quiescent (*p* = 3.65 × 10^−7^) and activated (*p* = 3.67 × 10^−3^) peaks. We then identified the closest gene to each peak containing a CLOCK motif, providing a list of putative CLOCK targets in quiescence and activation. Because of the high probability of false positives in this broad approach, we further refined this list by examining gene expression in CLOCKΔ19 cells, including only those genes which were 1.5-fold up- or downregulated in the mutant (Figure 6A).

Genes identified from quiescent peaks that were upregulated in the mutant, therefore likely repressed by WT protein, were significantly enriched for GO biological function “cell projection assembly”. Downregulated genes, likely activated by WT CLOCK, were significantly enriched for several metabolic processes associated with qHSC function (Figure 6C). Several of these genes were identified as targets in chromatin immunoprecipitation of CLOCK in liver [36]. This indicates that CLOCK may have a direct regulatory network which maintains a quiescent state in HSCs.

## 4. Discussion

The ability to arrest the progression of NAFLD into NASH, or predict individuals more likely to progress, would be a valuable tool against the increasing burden of chronic liver disease. Fibrosis underpins this progression, mediated primarily by HSCs depositing excess ECM. Our data show that disruption to circadian rhythm via CLOCKΔ19 mutation causes a primed phenotype in HSCs, leading to earlier activation of fibrosis-associated genes and significantly worse fibrotic response to injury. The pathways identified by our data may underpin the sensitivity of people with disrupted circadian rhythm to fibrotic injury and, thereby, progression to chronic liver disease.

Previous studies also correlated circadian rhythm disruption to more severe liver fibrosis. Our study complements those showing loss of Per2 exacerbating CCl_4_- and Bile Duct Ligation-induced injury [37]. Bmal1 was theorised to have a protective effect, with overexpression in the LX2 stellate model inhibiting glycolysis and the expression of activation markers in response to TGFB signalling. The authors also showed a reduction in Bmal1 in fibrotic tissue, HSCs from injured mice and TGFB activated LX2 cells [17]. In contrast, our data suggest Bmal1 does not alter significantly during in vitro HSC activation in aHSCs vs. qHSCs, regardless of CLOCK mutation status. This disparity may be due to our in vitro analysis focusing purely on the transition to activation without the dimension of injury. We also did not observe GO enrichment for glycolytic gene expression in CLOCKΔ19 HSCs vs. WT, suggesting altered glycolysis may be specific to modulation of Bmal1. One limitation of our in vitro experiments was that culture on plastic represents a mechanical stiffness many times higher than fibrotic liver tissue. Given the mechanosensitive nature of HSCs, it is likely that this non-physiological stiffness may cause an altered activation phenotype that masks differences between WT and CLOCKΔ19 HSCs. This could explain why the most pronounced phenotypic differences were observed in qHSCs directly extracted from the mice.

These previous studies were restricted to examining the phenotypes of activated or injured HSCs. To our knowledge, this study is the first to examine the pathways altered in quiescent HSCs with disrupted circadian rhythm to explore how this leads to a primed state for activation. Our study identified RHOGDI signalling as enhanced in CLOCKΔ19 qHSCs. RHOGDIs are known to bind various Rho GTPase proteins, acting to either activate or inhibit dependent on the specific GTPase and biological context [38] including spatiotemporal patterning [39]. Altered RHOGDI activity may, therefore, effect various cellular processes important to HSC activity including cytoskeletal organisation and migration. The complex interplay of GTPase signalling pathways may make it difficult to design therapeutic interventions, though components may still act as informative prognostic markers [40].

Though our ATAC data suggested a direct link between CLOCK and maintaining a qHSC state, further characterisation is needed to understand whether CLOCKΔ19 intrinsically primes HSCs towards a profibrotic state or whether the perturbation of metabolic pathways observed in CLOCKΔ19 mouse HSCs simply reflects the global phenotype of the animal. CLOCKΔ19 mice showed a metabolic syndrome-like phenotype including obesity, hyperglycaemia, and hyperlipidaemia. Additionally, CLOCKΔ19 mice showed disrupted intake patterns and hyperphagia [8]. With shorter fast periods between feeding, they may have increased susceptibility to disease as per studies exploring time-restricted feeding [41]. These underlying metabolic stressors may be sufficient to induce HSCs towards an active profibrotic state. In vitro disruption of CLOCK or other circadian genes in WT HSCs could discriminate a phenotype independent of the metabolic state of the animal. Nonetheless, characterisation of specific metabolic pathway changes correlating to this primed state in vivo may provide biomarkers for susceptibility to fibrosis and transitioning from NAFLD to NASH. Several of the genes highlighted by our analysis are already being assessed as biomarkers of disease severity [27,28,29,30,31,32,33,34,35].

## Figures and Tables

**Figure 1 cells-12-01582-f001:**
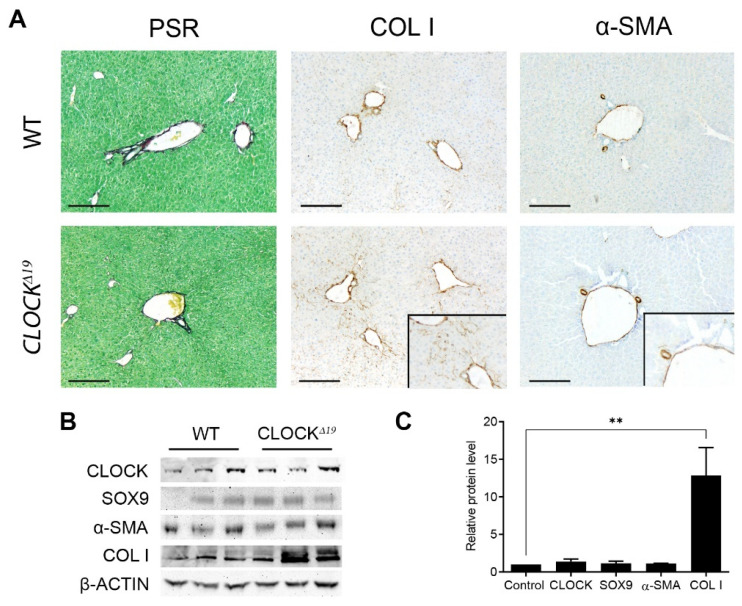
Naïve CLOCKΔ19 liver tissue shows enhanced COL1 deposition (**A**) IHC staining of WT and CLOCKΔ19 mouse liver tissue for fibrillar collagen (PSR), COL1, a-SMA and F4/80. COL1 in the CLOCKΔ19 mouse shows increased interstitial COL1 deposition (inset) with no evidence of interstitial a-SMA staining, suggesting a higher baseline COL1 deposition without an increase in activated stellate cells. (**B**) Western blot staining also shows significantly elevated COL 1 protein in CLOCKΔ19 mice, quantified in (**C**) but not in other fibrotic markers SOX9 and a-SMA (ordinary one-way ANOVA, ** = *p* < 0.01, *n* = 3). Error bars indicate mean ± SD.

**Figure 2 cells-12-01582-f002:**
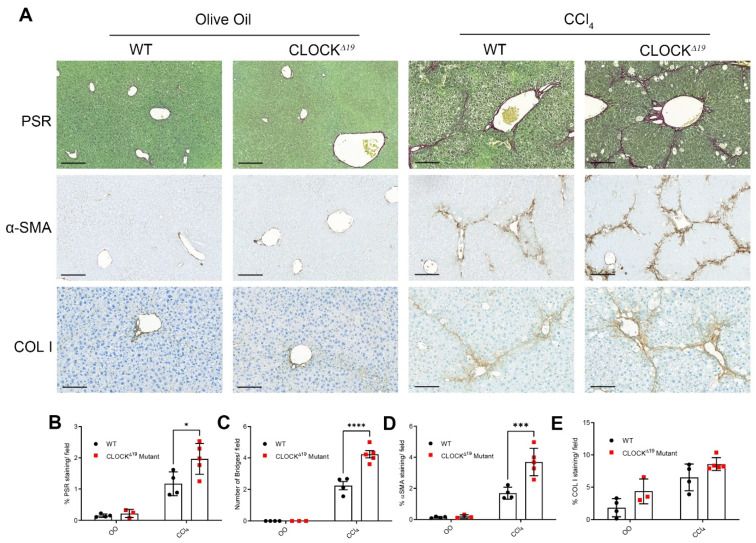
CLOCKΔ19 mice show an enhanced fibrotic response to CCl_4_ (**A**) Staining for fibrillar collagen (PSR) and IHC of a-SMA and COL1 in olive oil- and CCl_4_-treated mice shows enhanced fibrosis in the CLOCKΔ19 mutant. (**B**–**E**) Quantification of IHC showing significantly elevated (**B**) PSR staining, (**C**) bridging fibrosis and (**D**) a-sma expression in CCl_4_ treated mice. (**E**) COL1 shows a non-significant trend towards higher levels in the CLOCKΔ19 mice (two way ANOVA, * = *p* < 0.05, *** = *p* < 0.001, **** = *p* < 0.0001, *n* = 3–5). Error bars represent the mean ± SD.

**Figure 3 cells-12-01582-f003:**
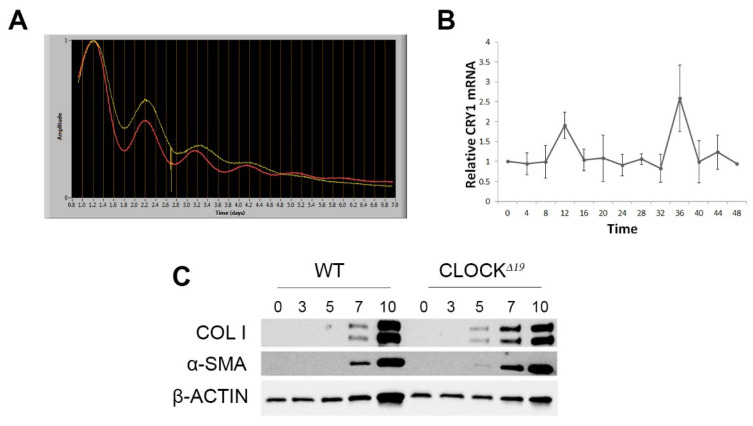
Rhythmicity in HSCs and accelerated in vitro activation of CLOCKΔ19 HSCs (**A**) Bioluminescence from PER2::Luc HSCs demonstrates circadian rhythm for around 4–5 days during in vitro activation. The dampening of amplitude over time is due to gradual desynchronisation in culture. Red and yellow traces represent two independent cell preparations. (**B**) RT-qPCR of Cry1 in HSCs from WT mice shows peaks of expression at 12 h and 36 h post-synchronisation (*n* = 3, error bars show mean ± SD) (**C**) Timecourse of COL1 and aSMA protein expression in HSCs during in vitro culture shows detectable levels at day 5 in CLOCKΔ19 HSCs, but not WT, indicating a more rapid activation of these cells.

**Figure 4 cells-12-01582-f004:**
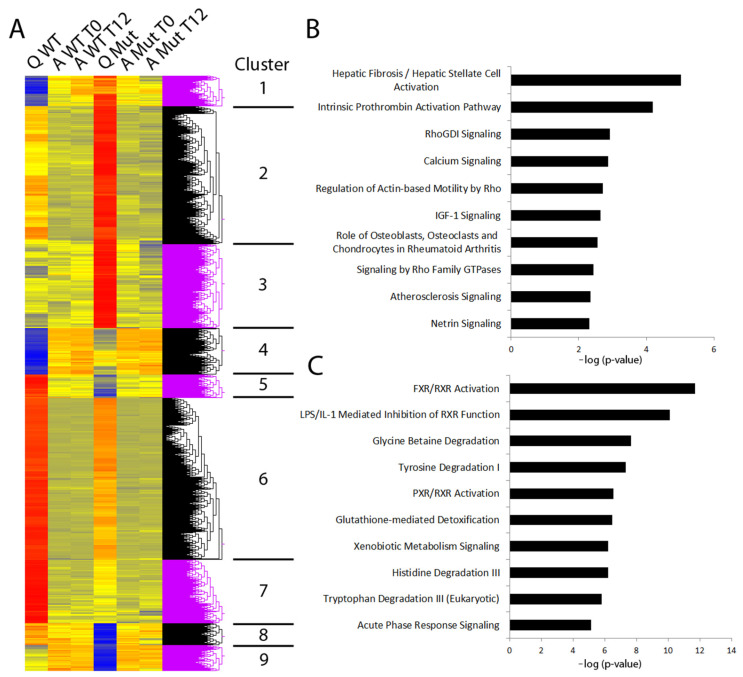
Microarray analysis of WT and CLOCKΔ19 HSCs (**A**) Gene clustering identified between WT and CLOCKΔ19 (Mut) HSCs in a quiescent state (Q), after activation in vitro and harvested immediately after (A T0) or 12 h post (A T12) synchronisation. (**B**) Top 10 pathways identified from genes in cluster 4 (genes positively correlating to activation and higher in Q Mut than Q WT) by Ingenuity Pathway Analysis includes HSC activation and RhoGDI signalling. (**C**) Top 10 pathways identified in cluster 6 (genes negatively correlating to activation and lower in Q Mut than Q WT) includes pathways involved in metabolic regulation.

**Figure 5 cells-12-01582-f005:**
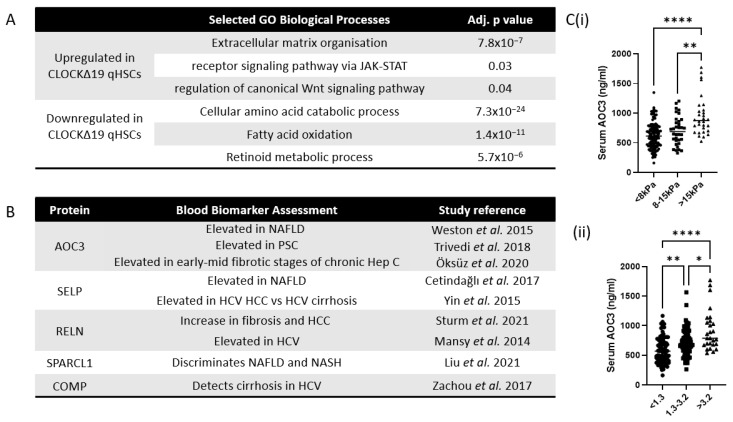
Genes upregulated by CLOCKΔ19 qHSCs include liver disease biomarkers (**A**) GO enrichment analysis of genes upregulated and downregulated in CLOCKΔ19 qHSCs (**B**) Selected genes upregulated in CLOCKΔ19 qHSCss predicted to reside in the extracellular space that have been assessed as non-invasive biomarkers [27,28,29,30,31,32,33,34,35]. (**C**) quantification of serum AOC3 in human patients of mixed liver disease aetiology grouped based on (**i**) liver stiffness as measured by transient elastography and (**ii**) FIB-4 score (Kruskal–Wallis test with Dunn’s multiple comparisons test * = *p* < 0.05, ** = *p*<0.01, **** = *p* < 0.0001).

**Figure 6 cells-12-01582-f006:**
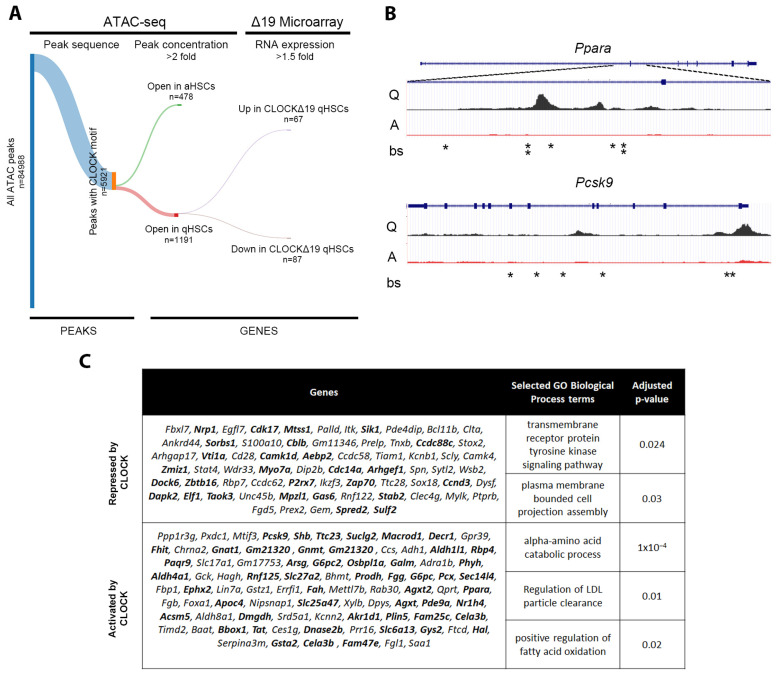
ATAC-seq integration with microarray data highlights a CLOCK regulome in qHSCs (**A**) ATAC-seq peaks containing a CLOCK TF binding motif were filtered according to 2-fold increased peak concentration in aHSCs/qHSCs and mapped to the nearest gene (Open in aHSCs) or similar for qHSCs/aHSCs (Open in qHSCs). This was then filtered according to the CLOCKΔ19 microarray expression data to include genes 1.5 fold up- or downregulated in CLOCKΔ19 mutant HSCs. Only genes associated with qHSCs emerged of which 67 were upregulated in CLOCKΔ19 qHSCs (thus modelled to be repressed by WT CLOCK) and 87 were downregulated (thus modelled to be activated by WT CLOCK). (**B**) Representative ATAC tracks for Ppara and Psck9 showing peaks in qHSCs (Q) and aHSCs (A). Asterisks indicate CLOCK binding sites (bs) (**C**) Table of genes modelled to be repressed or activated by CLOCK in qHSCs and associated GO Biological Processes. Bold gene names are ChIP validated CLOCK targets in liver based on [36].

## Data Availability

All data are within the paper and Appendix A. ATAC-seq analysis was based on data from [22].

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
