# Peer review of "Circadian Disruption Primes Myofibroblasts for Accelerated Activation as a Mechanism Underpinning Fibrotic Progression in Non-Alcoholic Fatty Liver Disease"

_cells, 2023, doi:10.3390/cells12121582_

Round 1

Reviewer 1 Report

This is a very interesting paper linking circadian clock function in liver stellate cells to NASH. They authors show that CLOCK regulates qHSC-to-aHSC transition and fibrosis. I have few comments:

1. Fig. 3 Show clock gene expression (Cry1 and others) in Clock-D19 HSCs.

2. Evaluate key candidate genes identified in Figs. 3 & 4 in synchronized HSCs.

3. Clock-D19 mice have a strong metabolic phenotype including hyperphagy and dirsupted intake patterns. These factors as potential contributors to liver fibrosis should be tested or at least discussed.

Author Response

We thank the reviewer for their constructive feedback, which we have used to strengthen our submission. Please find a point-by-point response below in red

This is a very interesting paper linking circadian clock function in liver stellate cells to NASH. They authors show that CLOCK regulates qHSC-to-aHSC transition and fibrosis. I have few comments:

  1. Fig. 3 Show clock gene expression (Cry1 and others) in Clock-D19 HSCs.
  2. Evaluate key candidate genes identified in Figs. 3 & 4 in synchronized HSCs.

Taking these two points together, we have extracted data for specific genes from our microarray to provide an overview of how the expression of various circadian rhythm genes are altered between the quiescent and activated state, between T0 and T12 synchronised aHSCs, and between the Clock-D19 and WT state. We provide this as supplementary table 3, discussing in line 269-273. We hope this provides readers an easily accessible assessment of key genes in the models we have used. We previously included the whole microarray output as a supplementary table, which would allow readers to similarly look for patterns in any given gene of interest.

  1. Clock-D19 mice have a strong metabolic phenotype including hyperphagy and dirsupted intake patterns. These factors as potential contributors to liver fibrosis should be tested or at least discussed.

Thank you. We have expanded our discussion to include the behavioural phenotypes displayed by these mice and how this might impact on a liver fibrosis phenotype (Lines 382-385).

Reviewer 2 Report

The study by author Jokl and colleagues shows how disruption of the circadian rhythm by its prolongation in such mice can lead to obesity, hyperglycemia, and hyperlipidemia, which can subsequently accelerate liver fibrosis that can cause the transition from NAFLD to NASH.

The methods are well described in the paper. The only thing that needs to be added is information about the genotype of the mice associated with CLOCK delta19. For example, are CLOCK delta19 mice homozygous or heterozygous for this mutation? The authors should provide this information because homozygous mice have a prolonged circadian rhythm of 4 hours and heterozygous mice of 1 hour. In addition, the data may be necessary for the interpretation of the results of this research.

The results are presented with the most critical research findings. However, the authors should show the results of human serum analysis in the paper because their analysis is mentioned in the Methods chapter. The authors should review these results in the context of this research and explain their relevance to circadian rhythm disorders.

The authors should show the limitations of this study.

A minor correction of the English language is needed - phrases such as 'in vitro' and 'in vivo' should be written in italics.

A minor correction of the English language is needed - phrases such as 'in vitro' and 'in vivo' should be written in italics.

Author Response

We thank the reviewer for their constructive feedback, which we have used to strengthen our submission. Please find a point-by-point response below in red.

The study by author Jokl and colleagues shows how disruption of the circadian rhythm by its prolongation in such mice can lead to obesity, hyperglycemia, and hyperlipidemia, which can subsequently accelerate liver fibrosis that can cause the transition from NAFLD to NASH.

The methods are well described in the paper. The only thing that needs to be added is information about the genotype of the mice associated with CLOCK delta19. For example, are CLOCK delta19 mice homozygous or heterozygous for this mutation? The authors should provide this information because homozygous mice have a prolonged circadian rhythm of 4 hours and heterozygous mice of 1 hour. In addition, the data may be necessary for the interpretation of the results of this research.

We thank the reviewer for highlighting this oversight. We have updated the methods to clarify that the mice used in our experiments were homozygous for the mutation (lines 100-101)

The results are presented with the most critical research findings. However, the authors should show the results of human serum analysis in the paper because their analysis is mentioned in the Methods chapter. The authors should review these results in the context of this research and explain their relevance to circadian rhythm disorders.

We believe the reviewer may have overlooked the serum analysis, which is present in figure 5C. We have set this analysis in the context of a biomarker for early detection of fibrosis in NAFLD – we cannot be confident whether AOC3 specifically has mechanistic relevance to circadian rhythm disruption per se, rather we view it as a potential early phenotype of liver fibrosis broadly given it has arisen as a biomarker for liver disease in multiple contexts.

The authors should show the limitations of this study.

We have discussed limitations throughout the discussion, as opposed to in a distinct section. We have incorporated some additional exploration of the limitations of our models throughout the discussion (eg lines 365-369).

 A minor correction of the English language is needed - phrases such as 'in vitro' and 'in vivo' should be written in italics.

Thank you for highlighting – this is now addressed.

Round 2

Reviewer 1 Report

The authors have addressed my concerns. I have no further questions.

n/a